# Effect of Recycled Polyethylene Terephthalate Strips on the Mechanical Properties of Cement-Treated Lateritic Sandy Soil

**Maitê Rocha Silveira [1], Paulo César Lodi [1,*], Natália de Souza Correia [2], Roger Augusto Rodrigues [1] and Heraldo Luiz Giacheti [1]**

1 Department of Civil and Environmental Engineering, São Paulo State University (UNESP), Av. Engenheiro Luiz Edmundo Carrijo Coube 14-01, Bauru, SP 17033-360, Brazil; maite81@hotmail.com (M.R.S.); roger.rodrigues@unesp.br (R.A.R.); h.giacheti@unesp.br (H.L.G.)

2 Department of Civil Engineering, Federal University of Sao Carlos (UFSCar), Rodovia Washington Luiz, São Carlos, SP 17033-360, Brazil; ncorreia@ufscar.br

* Correspondence: paulo.lodi@unesp.br; Tel.: +1-646-755-2239

**Abstract:** The civil engineering construction industry is nowadays one of the largest consumers of natural resources. Therefore, the proposal of using alternative materials that seek to reduce waste production or the use of previously generated waste is becoming increasingly necessary. This paper evaluated the effect of recycled polyethylene terephthalate (PET) strips on the mechanical properties of a cement-treated lateritic sandy soil. Unconfined compression strength (UCS) tests were conducted in natural and PET strips mixtures in different strips lengths and contents. In addition to UCS tests, compaction tests were also conducted in order to analyze the effect of these inclusions on the properties of a lateritic sandy soil. Lastly, direct shear tests were conducted on natural soil-strip, soil-cement, and soil-cement-strip composites using optimum UCS results. The addition of strips to the soil-cement composite showed an increase in the soil cohesion parameter. The inclusion of strips also provided a more ductile behavior to the soil, presenting greater deformations with fewer stress peaks. Results showed that the recycled strips' inclusion in soil-cement can provide a material with high strength, ductility, and a highly sustainable alternative.

**Keywords:** recycled pet strips; lateritic soil; cement; composite; uniaxial tests; shear strength

## 1. Introduction

Nowadays, the visible consequences of environmental degradation and the prediction of future environmentally catastrophic scenarios require drastic solutions for environmental conservation. Among them is the integrated management of solid wastes, in which a set of actions is proposed in order to promote sustainability and the preservation of natural resources through three main actions: reducing consumption, reusing consumed materials, and recycling generated waste.

The use of alternative materials in the civil engineering construction industry aiming to reduce the production of wastes or the use of previously generated wastes has become indispensable. In the context of the reduction of natural resource use in civil construction, such as soils, different materials appear as alternative options to compose soil-mixtures: synthetic or natural fibers, construction and demolition wastes, ashes or tire fibers. Another sustainable alternative is the addition of polymers, using previously generated wastes, such as plastic strips.

The study entitled "Fast facts about plastic pollution" by National Geographic written by Laura Parker [1] exposes the worrying situation of plastic in the world, showing the growing need for

recycling and reusing this material. According to the study, 40% of plastic produced is packaging, used only once and then discarded. Worldwide, 448 million tons of plastic have been produced since the beginning of this materials manufacturing, of which 44% has been made since 2000. Currently, less than a fifth of all plastic is globally recycled. In Europe, plastic recycling rates are higher than 30 percent, while in China, the rate of plastic recycling is 25 percent. In the United States, plastic recycling is only 9 percent of total plastic trash.

After numerous studies indicating a high potential for the application of polymeric fiber reinforced-composites in improving the mechanical properties of soils, recent studies started to evaluate the influence of using polymeric strips as soil reinforcement materials, e.g., References [2–6].

The main difference between the two types of inclusions is in the shape of the materials. While polymeric fibers are very thin and elongated materials, such as filaments, polymeric strips are materials of greater width and thickness, usually cut from existing plastic structures. The use of Polyethylene Terephthalate (PET) strips as soil improvement has several advantages, such as the possibility of reusing plastic waste to increase soil strength without the need of a recycling process, as in the case of synthetic fibers.

Sivakumar Babu and Chouksey [2] evaluated the effect of including strips 12 mm long and 4 mm wide, in quantities of 0.50%, 0.75% and 1.0% in a sandy soil through unconfined strength tests and triaxial tests (consolidated and not drained). The authors noted significant increases in the soil shear strength parameters (cohesion and internal friction angle), which were greater for greater amounts of strips added. In addition, unconfined strength tests indicated an increase in ductility, proportional to the inclusion of strips. Soltani-Jigheh [3] studied the inclusion of plastic strips 4 mm wide and 8 mm long in quantities of 0.25; 0.50; 0.75; 1.0; 1.5 and 2% in relation to the mass of a clayey soil by performing triaxial tests (consolidated and not drained). The results showed a small increase in the shear strength of the soil. In general, changes obtained in the shear strength of soils were also small, resulting, in general, in an increase of cohesion and decrease of the friction angle.

Studies have also addressed the use of cement or lime to improve soil properties. The large number of studies that evaluated the use of soil-cement mixtures may be justified by the characteristics obtained with this composite, which presents a significant increase in natural soil strength and stiffness [6–17]. Although cement is considered a high-environmental-impact material, these studies aim to obtain a material with high strength and durability from the inclusion of small amounts of cement, reducing the use of other polluting materials. Specht et al. [9] suggest that a cement-treated soil usually shows an increase in soil strength and stiffness, turning this mixture into an ideal material for several geotechnical applications, such as the base of shallow foundations, slope protection and base/sub-base of flexible pavements. However, great brittleness and high cracking potential have discouraged the use of such material in pavement engineering. In this sense, the addition of fibers and strips in a soil-cement composite can provide a material with high strength and more ductile behavior.

Specht [9] studied the behavior of soil-cement-fiber mixtures submitted to static and dynamic loads aiming at paving. The author stated that the influence of the inclusion of polypropylene fibers on the properties of the composite essentially depends on the fiber and matrix. Still, according to Specht [9], more flexible fibers showed a more pronounced effect on post-peak behavior, increasing ductility, toughness, and resistance to fatigue, while fibers with greater stiffness pronounced the effect of increasing peak strength. In addition, longer fibers were more effective.

Guedes [14] analyzed the mechanical performance of a soil-cement micro reinforced with synthetic polypropylene fibers for use as a primary coating on unpaved roads. The study incorporated fibers of 6 and 24 mm lengths into the soil-cement in proportions of 0.25%, 0.50%, and 0.75%. The incorporation of fibers into the composite proved to be satisfactory, increasing the peak strength and the drop in strength after the peak. Other factors that were significantly affected by the inclusion are the increase in break deformations, decrease in stiffness, increase in elastic and plastic deformations, and reduction in deformability modules and strength to compaction. The 24 mm fiber incorporated into the soil at 0.75% content proved to be the most influential combination.

Girardello [15] studied the behavior of pullout tests of embedded plates in soil-cement-fiber layers using polymeric polypropylene fibers. The results indicated an increase in the strength required for the removal of the embedded sand-cement, sand-cement-fiber, and sand-fiber plates when compared to the removal of the embedded sand plates. Changes were observed in the form of soil rupture when reinforced with fibers and/or cement.

Cristelo et al. [12] analyzed microscopic images of the soil in its pure state, with the addition of fibers, cement, and with the addition of cement and fibers together. The authors observed the soil structure and concluded that there is an increase in the void index with the addition of fibers, which are responsible for a loss of mechanical strength, resulting from the friction between the particles. The addition of cement, unlike what occurs with the addition of fibers, is responsible for decreasing the void index of the composite, since it acts as a binder. In the case of the mixture of soil, cement and fibers, cement acts by reducing voids and increasing the bond strength at the soil-fiber interface, providing improvement in the mechanical strength of the mixture.

Regarding the inclusion of strips, Olutaiwo and Ezegbunem [16] evaluated the effect of cement and PET bottle strips in a lateritic soil using Modified Proctor and CBR tests. The amount of cement varied between 0%, 1%, 3%, 5%, and 7%, and the number of strips (5 mm wide and 10 mm long) in 0%, 5%, 10%, 15% and 20%. Results showed a decrease in the optimum humidity and an increase in the maximum dry density, which were shown to be greater in the inclusion of 10% of strips for all the cement quantities. As for the CBR tests, all cement-strip combinations evaluated increased the soil's bearing capacity. In all cement inclusions, the inclusion of 10% of strips generated the greatest increase in strength. The most effective inclusion was 7% cement with 10% strips, generating a 326.8% increase in soil support capacity. Overall, the addition of the strips to the soil-cement presented a beneficial alternative to the environment, in addition to being economical when compared to the single addition of cement.

Tang et al. [17] evaluated the inclusion of natural fiber reinforcements within cemented soils and found an increase in UCS and changes on the brittle behavior of cemented soil to a more ductile behavior. Olgun et al. [18] evaluated the effect of polypropylene (PP) fibers inclusion on the strength of cement-fly ash stabilized clay soil and found that the main advantage of fiber reinforcement was the improvement in material ductility, particularly above 0.5% fiber content, and with increased fibers length.

Therefore, the present study aims to expand the understanding of the shear strength and deformability behavior of a cement-treated lateritic soil mixture with recycled polymeric strips. Considering the high cost of the cement and the polluting potential of these materials, the recycled strips were added to a cement-treated soil seeking a low-cost material with high strength, ductility and a highly sustainable alternative. This practice seeks to assess how much strength can increase with the addition of the recycled strips in order to maximize their use and minimize the use of cement. Unconfined compression and direct shear tests were conducted using soil-strip, soil-cement, and soil-strip-cemented mixtures. A lateritic sandy soil was used in the present study for the evaluation of the effect of recycled strips and cement inclusion in soils found in tropical zones.

## 2. Materials and Methods

Sandy soil samples were collected from the experimental campus of the São Paulo State University (UNESP) at Bauru, in Sao Paulo State, Brazil. Soil samples were characterized according the following recommendations: particle size analysis—ABNT NBR 7181 [19], liquidity limit—ABNT NBR 6459 [20], plasticity limit—ABNT NBR 7180 [21], specific density of solids—ABNT NBR 6458 [22], unconfined compression strength—ABNT NBR 6457 [23] and ABNT 7182 [24], and direct shear—ASTM D3080 [25]. The soil was classified as a medium to fine, reddish brown, clayey sand, according to the classification adopted by ABNT NBR 6502 [26]. These materials are residual soils formed in humid tropical regions with a predominance of weathering. Lateritic soils are characterized in their formation by the intense migration of particles under the action of infiltrations and evaporations, giving rise to a porous surface

horizon, remaining almost exclusively the most stable minerals (quartz, magnetite, illite and kaolinite). In these soils, the presence of aggregated clay and silt particles is common owing to the action of iron and aluminum oxides and hydroxides, which gives these soils characteristics of mechanical and hydraulic behavior not consistent with their texture. Lateritic fine-grained soils have superior properties when compacted; however, they can show unfavorable properties such as cracks, shrinkage, water sensitivity and irregular distribution [27–30]. Table 1 illustrates soil properties obtained in tests for characterization of this soil.

**Table 1.** Properties of the lateritic soil.

| Parameter | Unit | Value |
|---|---|---|
| Sand | % | 80.2 |
| Silt | % | 5.8 |
| Clay | % | 14.0 |
| Liquid Limit | % | 16.0 |
| Plastic Limit | % | NP * |
| Maximum dry density, $\rho d_{máx}$ | g/cm$^3$ | 1.950 |
| Optimal moisture content, $w_{opt}$ | % | 10.6 |
| Specific density of solids, $\rho_s$ | - | 2.649 |

* NP = Non-Plastic.

The Portland cement used in this research is type CP II-F-32, manufactured by CSN (Brazil). This cement was chosen, according to Brazilian Portland Cement Association (ABCP), due to its wide availability in Brazil. The cement was added to soil in percentages of 2, 4, 6, 8 and 10%.

Plastic wastes used in this research were composed by PET bottle. Strips were cut from Coca-Cola® plastic bottles and randomly distributed in the soil. Bottles were previously sanitized under running water. After this process, a portion of the bottle in which the label was placed was cut and separated from the rest of the packaging to form the strip sample. The sample was cut into strips with 1.5 mm width by 10, 15, 20, and 30 mm length. The percentual of strips (0.0, 0.25, 0.5, 0.75, 1.0, 1.5, and 2%) was added to soils in relation to the mass of dry soil (Figure 1). The PET strips have a specific mass of 1.30 g/cm$^3$, a tensile strength of 1000 MPa, and a tensile modulus of 15.0 GPa. The aspect ratio (AR) for the strips having a length of 10 mm, 15 mm, 20 mm, and 30 mm are 20, 30, 40, and 60, respectively. The cutting process of the PET strips, the final shape of the strips, and an example of soil mixed with strips are shown in Figure 1.

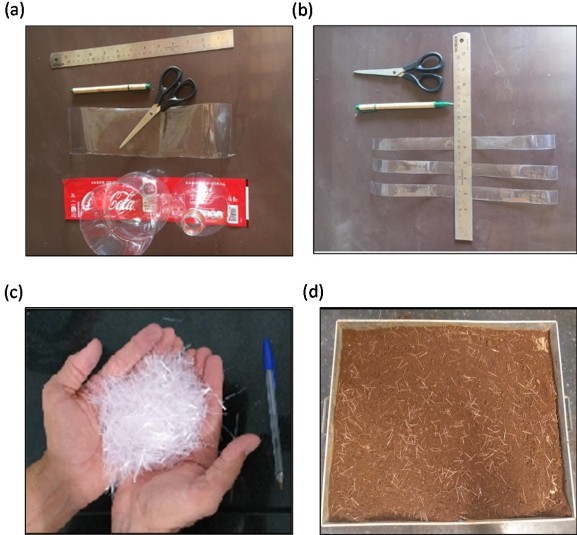

**Figure 1.** PET strips: (**a**) PET samples; (**b**) the cutting process; (**c**) PET strips after cutting; (**d**) soil mixed with PET strips.

Standard Proctor tests—ABNT NBR 7182 [24] were conducted with the addition of strips to the soil in order to evaluate the effect of including different sizes and percentages of strips to the soil compaction parameters (maximum dry density and optimum moisture content). Proctor tests with the addition of cement—ABNT NBR 12,023 [31] were also conducted in percentages of 2% and 10% to obtain the optimal parameters for the soil-cement mixture.

Unconfined compression tests—ABNT NBR 12,770 [32] were also conducted for natural soil and soil-strip mixtures. Due to the small variations obtained with the inclusion of strips in the compaction parameters (see item 3.1.1), a natural soil compaction curve was adopted for the molding of the specimens, which were compacted in the optimum moisture (±0.5) and with a 100% degree of compaction, with an acceptable variation of 3%. UCS tests were performed with 3 specimens for each length and respective strips content, totalizing 24 unconfined compression strength tests, allowing an analysis of the effect of the inclusion of strips in the soil in terms of strength improvement and variability.

The direct shear tests were conducted according to ASTM D3080 [25] with the addition of cement to soil in percentages of 2, 4, 6, 8, and 10%. In addition to the soil-cement tests, soil-cement-strips tests were also performed. The direct shear tests were performed with the molding of the specimens in the optimum soil content, using a previously calculated mass of soil or soil-cement according to the volume of each mold and the maximum density of the composite. The sought degree of compaction was 95% for the tests with the soil-cement and soil-cement-strip. For the tests with the soil-strip the degree of compaction of 100% was adopted for comparison with the unconfined compression test (UCS) results. The static compaction procedure was adopted, in which a metal plunger was attached to the same simple compression equipment and introduced into the metal mold of direct shear test at a constant speed, compacting the amount of soil, strip, and water previously calculated, weighed with a resolution of 0.01 gf, and homogenized to obtain maximum dry density and optimal moisture content according to compaction degree (Figure 2a,b). The specimens were then placed inside a capsule that was attached to the direct shear machine to initiate the shearing (Figure 2c,d). The specimens were densified with the use of loads of 1, 2 and 4 kg, which resulted in normal stresses of 30.56, 61.11, and 122.22 kPa, respectively. In the execution of tests with the use of soil-cement and soil-cement-strip composites, curing was carried out by placing the specimens in a humid chamber, with constant wetting for periods of 7, 14, and 28 days. The tests were carried out with the specimens of 7 days due to similarity with the results obtained considering the periods of 14 and 28 days.

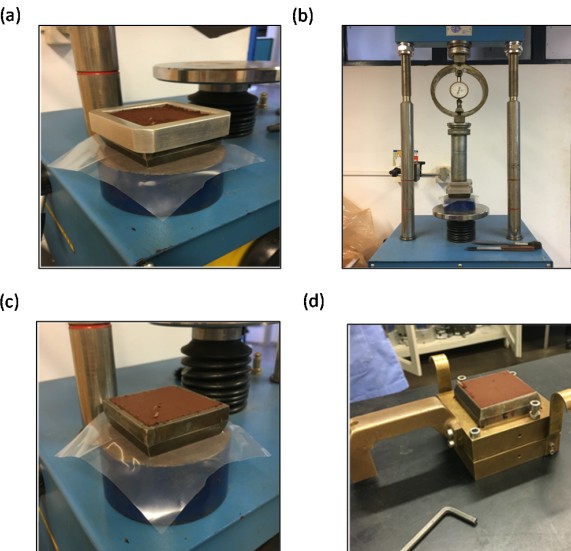

**Figure 2.** Molding of the specimens for the direct shear test. (**a**) Placing the soil on the mold; (**b**) compacting the soil using the machine; (**c**) the molded specimen; (**d**) placement of the specimen inside the capsule.

Considering the high cost of cement and the potential pollutant of the material, the strips were added to the sandy matrix composites with 2% cement, looking for a low-cost material, with high strength, ductility and high sustainable potential, seeking to assess how much the strength could increase with the addition of the strips so to maximize their use and minimize the use of cement. The analysis of the results was based on the analysis of the shear strength soil parameters (cohesion and internal friction angle) and the stress-displacement curves taking into account the soil, the soil-cement, and the soil-cement-strip.

## 3. Results and Discussion

### 3.1. Compaction Tests

#### 3.1.1. Compaction Tests in Soil-Strip Mixtures

Table 2 shows the optimal compaction parameters obtained for soil-strip composites, with strip inclusions in different sizes and percentages.

**Table 2.** Proctor test results performed (soil-strip).

| Strip Length (mm) | Inclusion (%) | Optimal Moisture Content (%) | Maximum Dry Density (g/cm$^3$) |
|---|---|---|---|
| 10 | 0.25 | 10.4 | 1.940 |
|  | 0.50 | 9.4 | 1.945 |
|  | 0.75 | 10.1 | 1.915 |
|  | 1.00 | 10.1 | 1.905 |
|  | 1.50 | 10,4 | 1.915 |
|  | 2.00 | 10.9 | 1.905 |
| 15 | 0.25 | 10.4 | 1.925 |
|  | 0.50 | 10.2 | 1.925 |
|  | 0.75 | 10.4 | 1.925 |
|  | 1.00 | 10.4 | 1,930 |
|  | 1.50 | 10.7 | 1.900 |
|  | 2.00 | 10.9 | 1.905 |
| 20 | 0.25 | 10.3 | 1.945 |
|  | 0.50 | 10.8 | 1.900 |
|  | 0.75 | 9.6 | 1.925 |
|  | 1.00 | 10.5 | 1.940 |
|  | 1.50 | 10.8 | 1.895 |
|  | 2.00 | 10.7 | 1.880 |
| 30 | 0.25 | 10.2 | 1.940 |
|  | 0.50 | 10.9 | 1.940 |
|  | 0.75 | 9.9 | 1.925 |
|  | 1.00 | 10.6 | 1.930 |
|  | 1.50 | 10.8 | 1.905 |
|  | 2.00 | 10.6 | 1.880 |

In all cases analyzed it should be noted that there is a decrease in the maximum dry density obtained due to the inclusion of the strips regarding the results with the optimal parameters obtained from compaction test on natural lateritic sandy soil (Table 1). The decrease in the maximum dry density ranged from 3.58% to 0.26%. The highest specific mass was obtained for inclusions of strips in lower percentages (0.5% inclusions of 10 mm long strips and 0.25% 20 mm long strips). The lowest values were obtained for inclusions of longer length strips and in the highest percentages evaluated (20 and 30 mm long strips in additions of 2.0% in relation to the dry mass of the soil).

Given that the specific mass of PET (1.30 g/cm$^3$) is less than the maximum dry density of the analyzed soil (1.95 g/cm$^3$), it is not possible to conclude whether the addition of strips confers strength to compaction and increases the porosity of the mixture, as stated by Hoare [33] and Festugato et al. [34],

or if it occurs due to the addition of a reduced specific mass material. Future evaluation of soil and soil-strip permeability is indicated to obtain more specific results.

### 3.1.2. Compaction Tests in Cement-Soil Composites

Table 3 shows the compaction parameters obtained for the natural lateritic sandy soil and with a 2% and 10% cement addition. The addition of cement to the soil showed no relevant variations in the soil compaction parameters, culminating in a very small variation in the maximum dry density and in a small reduction of de optimum moisture content.

**Table 3.** Proctor test results performed (soil-cement).

| Cement Addition (%) | Optimal Moisture Content (%) | Maximum Dry Density (g/cm$^3$) |
|---|---|---|
| 0.0 | 10.6 | 1.950 |
| 2.0 | 10.3 | 1.940 |
| 10.0 | 10.0 | 1.960 |

### 3.2. Unconfined Compressive Tests (UCS)

The medium values of unconfined strength (average from three tests for each combination of length and strip content) obtained by the inclusion of PET strips in the soil are shown in Table 4 and Figure 3.

**Table 4.** Values of unconfined compressive strength (kPa) (PET strip content—lateritic sandy soil).

| | Length (mm) | | | |
|---|---|---|---|---|
| | **10** | **15** | **20** | **30** |
| **(%)** | | | | |
| 0 | 56.72 | 56.72 | 56.72 | 56.72 |
| 0.25 | 73.68 | 78.92 | 81.07 | 84.29 |
| 0.50 | 79.83 | 86.29 | 93.19 | 88.82 |
| 0.75 | 86.41 | 87.81 | 99.07 | 95.86 |
| 1.0 | 90.49 | 95.98 | 105.76 | 99.73 |
| 1.5 | 98.02 | 100.72 | 109.34 * | 102.10 |
| 2.0 | 91.89 | 97.99 | 103.61 | 99.54 |

\* Highest value obtained.

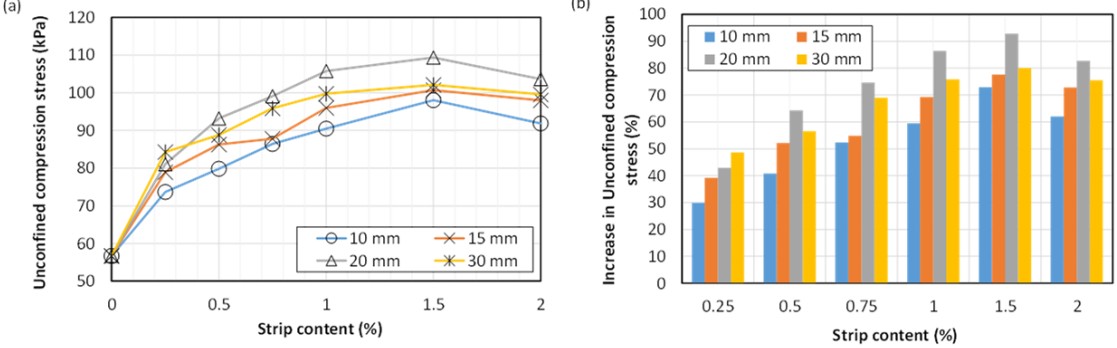

**Figure 3.** UCS results of lateritic sandy soil/PET strips. (**a**) UCS with fiber content increase; (**b**) the increase in UCS.

Results show that the unconfined strength of the samples varies for all sizes and percentages of added strips. In all analyzed cases there is an increase in strength due to the inclusion of strips, regardless of the length and percentages in which they were included. The optimum soil-strip

parameter, that is, the inclusion of strips in which the size and percentage resulted in greater strength increase, was found for the inclusion of strips with 20 mm in length and 1.5% content in relation to the dry mass of the soil. The result is in an increase of 92.4% in relation to the strength of the natural soil.

Analyzing the behavior of the soil (Figure 3) for different types of strips (either in length or quantity), there is a tendency to increase soil strength the longer the length of the added strips.

Figure 4 shows the stress-strain curves obtained in the test considering the natural lateritic sandy soil and the lateritic sandy soil with the inclusion of the optimum soil-strip parameter (strips 20 mm long and 1.5% in relation to the dry mass of the soil). It is noted that the inclusion of this strip parameter not only increases the load capacity of the soil, but also the ductility, increasing the deformation of the soil before rupture by about 1.7%.

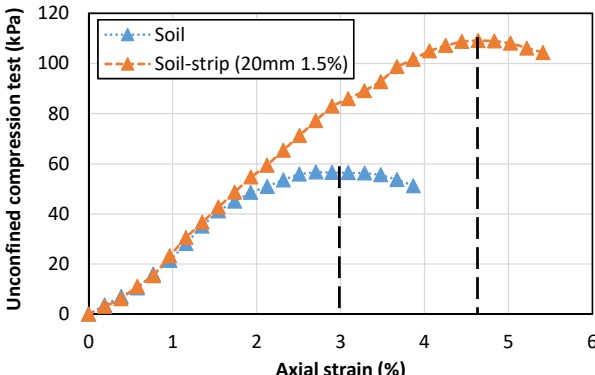

**Figure 4.** Stress-strain curves obtained: natural lateritic sandy soil and lateritic sandy soil with the inclusion of the optimum parameter soil-strips (strips 20 mm long and 1.5% content).

The strength of strip-reinforced soil increases with the increasing aspect ratio ($A_R$) of fibers. These results are in accordance with the literature, e.g., Shukla [5]. The shear strength of soil increased with increased PET content and the use of strips contributed to change soil brittle failure into ductile failure [2,35]. As discussed by Tang et al. [36], the increase in UCS might be related to the bridging effect of the fiber, which can efficiently avoid the later development of failure planes and strains in the soil.

### *3.3. Direct shear tests*

#### 3.3.1. Direct Shear Tests in Soil-Strip Composites

Direct shear tests were conducted on lateritic sandy soil (with a 100% degree of compaction) and on soil with the optimal soil-strip parameter inclusion (1.5% inclusions of 20 mm long strips obtained from UCS) (Table 5). The inclusion of strips in the evaluated parameter was effective in increasing the shear strength of the soil for normal stresses smaller than 300 kPa, presenting an increase of 66.7% in cohesion and a decrease of 3.5% in the friction angle in relation to the parameters obtained for the soil without the inclusion of strips.

**Table 5.** Shear strength parameters of soils obtained in direct shear tests for lateritic sandy soil with and without the inclusion of strips.

| Sample | c (kPa) | φ (º) |
|---|---|---|
| Soil with 0% strips | 11.7 | 31.4 |
| Soil with 1.5% (L = 20 mm) | 19.5 | 30.3 |

c = cohesion; φ = effective friction angle.

Figure 5 shows the shear stress-displacement curves for both soils (with and without strips). These curves are very similar considering the highest applied stresses and slightly more distant from each other considering the lowest stresses.

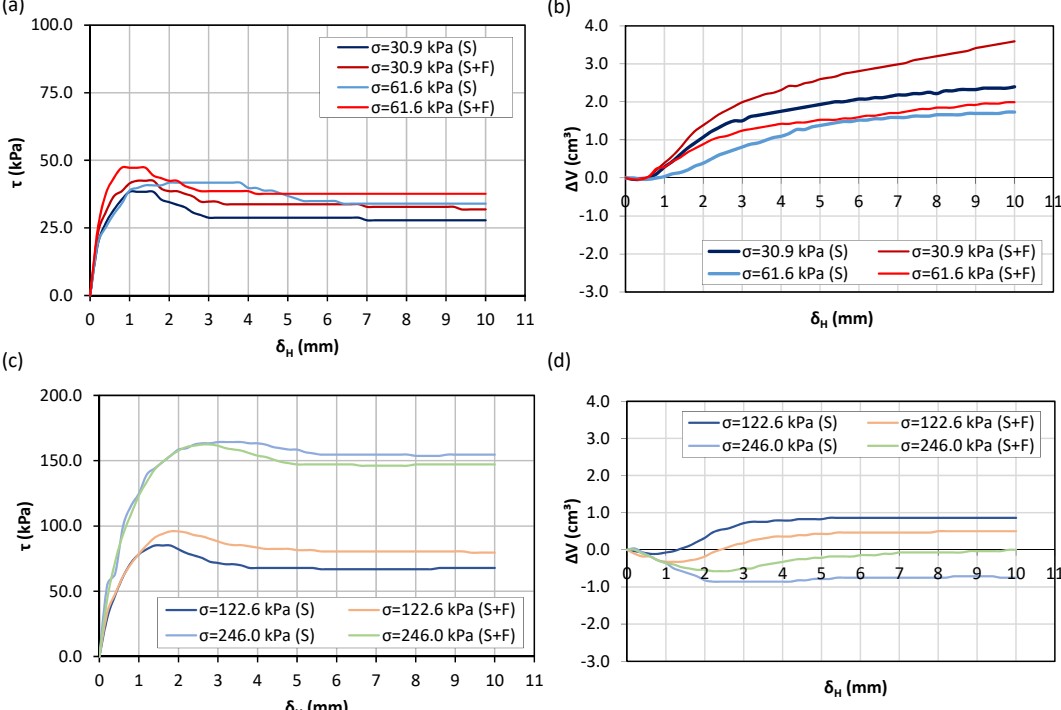

**Figure 5.** Direct shear test results: 1.5% content of 20 mm PET strips (S+F) and without strips (S). (**a**) shear stress-displacement (lower normal stress); (**b**) volume-displacement (lower normal stress); (**c**) shear stress-displacement (higher normal stress); (**d**) volume-displacement (higher normal stress);

Considering the volume-displacement curves, an increase in volume variation on the lateritic sandy soil at the shearing (dilatancy) was observed when lower normal stresses were applied. The soils with strips presented a slightly higher variation when compared to soil without strips. For higher normal stresses, there is a trend of decrease in volume variation. The volumetric variation showed that the soil presented a fragile rupture (with dilation) and the addition of the strips modified the behavior of the soil for the highest applied stress levels (the material starts to present a ductile behavior). Maher and Gray [37] and Consoli et al. [10] suggest that the effect of fiber inclusion on dilation and volume change is pronounced at higher load and strain levels, possibly due to the inhibiting action of the fibers. Many authors have reported that the addition of fibers tends to increase the ductility and strength of the soil-fiber composite as well, e.g., References [6,34,38–41].

### 3.3.2. Direct Shear Tests in Soil-Cement Composites

The tests using soil-cement and soil-cement-strip composites were performed with a 95% degree of compaction. Table 6 shows the shear strength parameters of the soil and the soil-cement composites found for each shear strength envelope, and Figure 6 presents the shear strength envelopes with and without inclusions of cement for comparison purposes.

**Table 6.** Shear strength parameters obtained for the soil with cement content.

| Cement Addition (%) | c (kPa) | φ (°) |
|:---:|:---:|:---:|
| 0 | 6.2 | 31.9 |
| 2 | 23.7 | 46.5 |
| 4 | 116.5 | 48.1 |
| 6 | 60.2 | 56.5 |
| 8 | 154.2 | 56.2 |
| 10 | 95.2 | 76.5 |

c = cohesion; φ = effective friction angle.

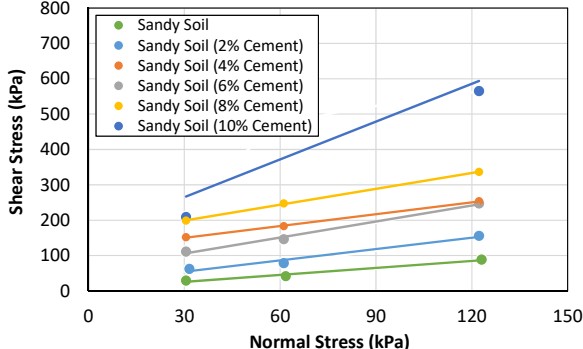

**Figure 6.** Failure envelopes with the cement addition.

In all the cases evaluated herein, the addition of cement was mostly effective in increasing soil shear strength via the increase in both the cohesion and friction angle parameters in large proportions, even in cases where small contents of cement were added. The reinforcement of the soil with the use of cement had its main effect in the creation of a cohesive intercept in the composite, making its application interesting in granular soils. Most of the shear strength of this type of matrix is due to friction between the particles, which also showed considerable increases. Regarding the increase in cohesion, the most pronounced effects were noted for additions of 8% and 4% of cement, while the friction angle shows larger changes for additions 10% and 6% of cement editions.

Figure 7 shows the stress-displacement curves obtained in direct shear tests of soil and soil-cement. Analyzing the curves, it is possible to notice that the rupture of the specimens with the addition of cement occurs after greater displacement than that of the soil without the cement addition. However, there is a trend for greater loss of strength after the peak. This behavior is clearly noticeable for inclusions of 4% and 6% of cement and accentuated for inclusions of 8% and 10% of cement. Thus, it can be concluded that the addition of cement alters the rupture of the material, making the rupture fragile so that the material strength decreases sharply as the deformation increases. In materials that present this type of rupture, the collapse process can be very fast, generating catastrophic situations.

These results obtained herein are in accordance with the literature [11–13,38]. Also, the results are in agreement of the conclusions of Festugato et al. [34] and Consoli [42], in which the use of cement percentages greater than 5%, in relation to the dry weight of the soil, gave a more significant stiffness to the composite. The addition of percentages of the order of 3% showed partial improvement in the matrix properties, mainly related to workability, with a certain increase in the carrying capacity.

In this sense, a good alternative to avoid this behavior is the addition of strips in soil-cement composite. In order to evaluate the influence of the strips on the strength, ductility, and mainly on the residual (post-peak) strength of the soil-cement composites, a new composite was tested: the soil-cement-strip. The tests on this composite were performed with the inclusion of PET strips in different lengths (10, 15, 20, and 30 mm) and percentages (0.75; 1.0; 1.5, and 2.0%—item 3.3.3). As previously mentioned, considering the high cost of the cement and the polluting potential of these materials, the strips were added only to a soil-cement with a 2% cement addition, seeking a low-cost

material with high strength, ductility, and a highly sustainable alternative. This expedient seeks to assess how much strength can be increased with the addition of the strips in order to maximize their use and minimize the use of cement.

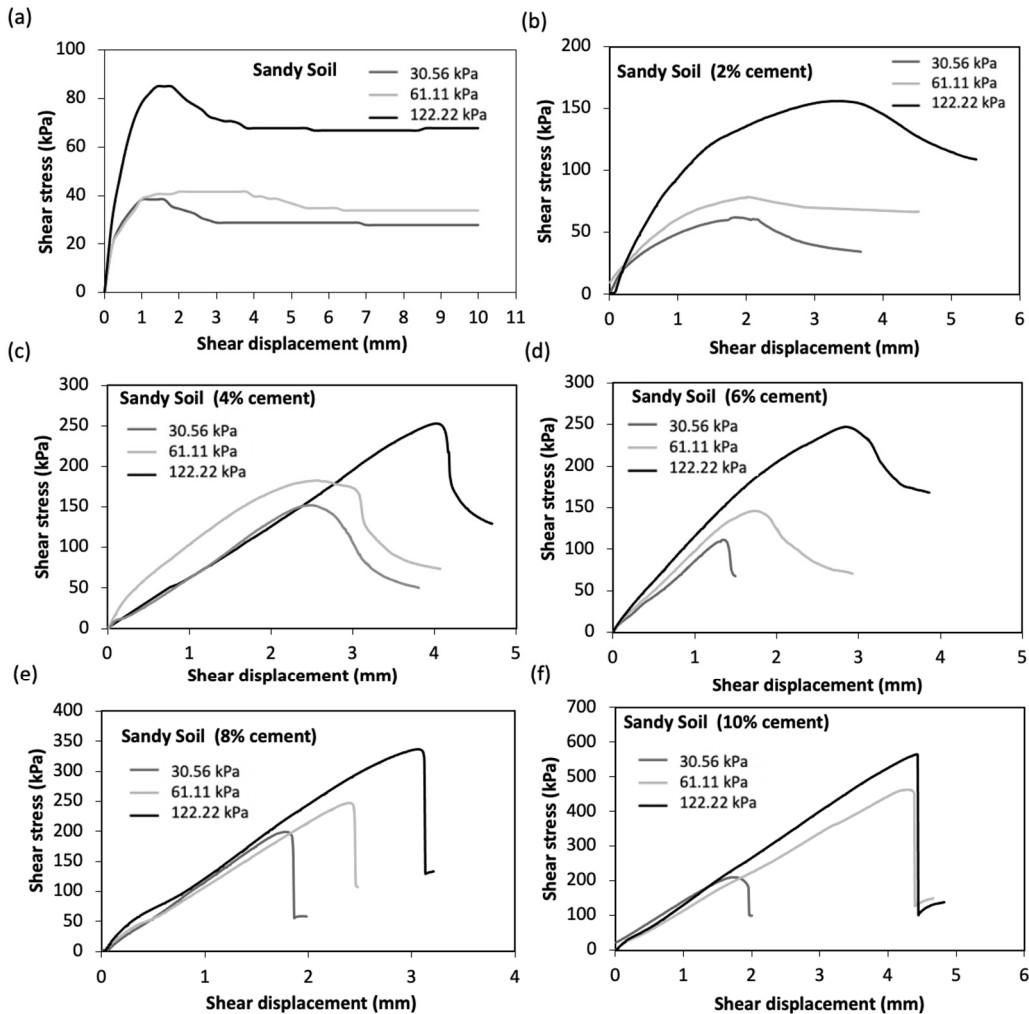

**Figure 7.** Stress-displacement curves obtained in soil and soil-cement.

### 3.3.3. Direct Shear Tests in Soil-Cement-Strip Composite

Figures 8–11 show the shear stress-displacement curves regarding the soil-cement-strip composite. Results from Figures 8–11 showed that the inclusion of strips provides, in general, a ductile behavior to the material, presenting greater deformations and lower stress peaks, in comparison with the data obtained in tests carried out with the addition of 2% cement to the lateritic sandy soil. This behavior is more pronounced for inclusions of larger strips, 30 mm long, and in larger quantities. The inclusion of 10 mm strips had a greater effect in decreasing the drop in the post-peak strength of the material, presenting deformations at peak slightly lower than those obtained for soil-cement with the addition of 2% cement. The inclusion of strips of 15 and 20 mm presented greater deformations of the material; however, the drops in strength after the peak were more pronounced in these types of inclusions. Finally, in relation to the stress-displacement behavior of the material, the inclusion of 30 mm strips in amounts of 1.5 and 2.0% proved to be more effective, presenting both greater deformations and lower post-peak strength drops compared to the data obtained in the tests with the addition of 2% cement to the soil.

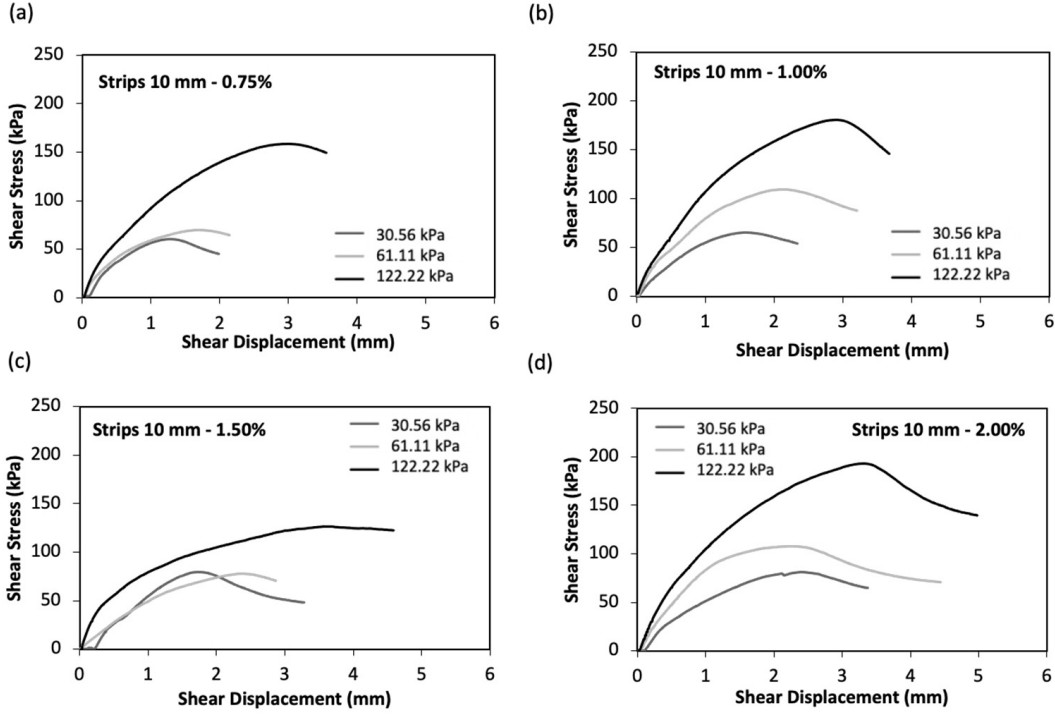

**Figure 8.** Direct shear results: the soil-cement-strip (2% cement—L = 10 mm).

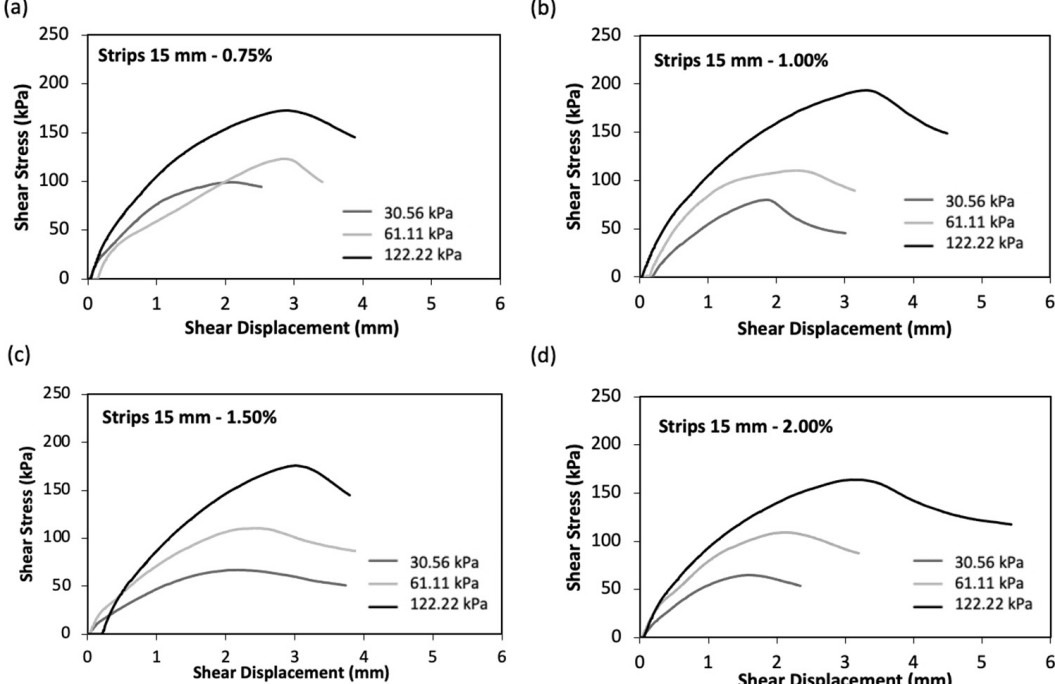

**Figure 9.** Direct shear results: the soil-cement-strip (2% cement—L = 15 mm).

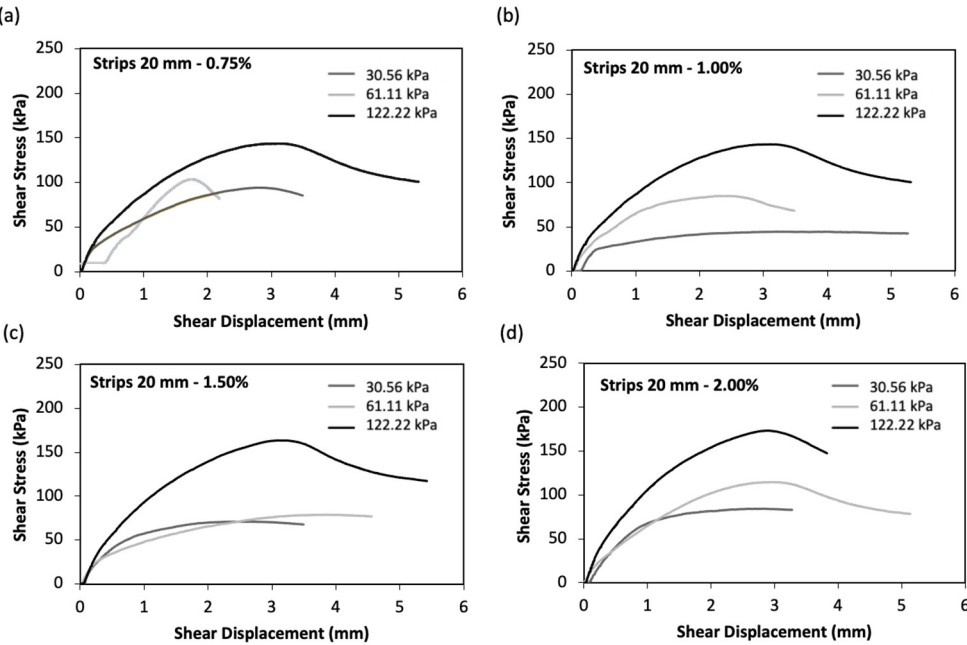

**Figure 10.** Direct shear results: the soil-cement-strip (2% cement—L = 20 mm).

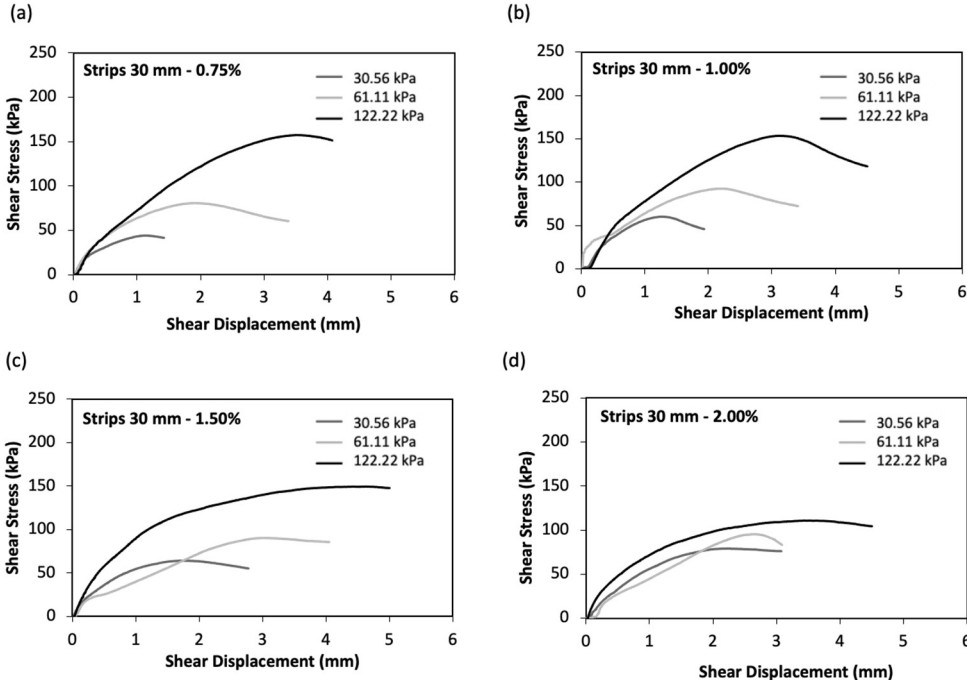

**Figure 11.** Direct shear results: the soil-cement-strip (2% cement—L = 30 mm).

The benefits of the addition of strips in increasing the ductility of the composite can be seen in terms of the variation of the secant elasticity modulus ($E_{ps}$). The secant modulus of elasticity was calculated from the strain at peak strength ($E_{ps}$).

The values presented in Table 7 and Figure 12 show the variation of $E_{ps}$ with strip content and cement content. It can be noted that the secant modulus decreases with the increase of the strip content as well as with the increase of the cement content. This is because extensible fibers require an initial deformation to initiate strength mobilization, resulting in the reduction of fiber-reinforced cemented soil stiffness. Moreover, the addition of higher fiber contents into the cemented soil matrix may lead to a significant drop in stiffness [43].

**Table 7.** Secant modulus at peak strength for soil-cement-strip with 2% cement content.

| Content (%) | L (mm) | | | | |
|---|---|---|---|---|---|
| | 0 | 10 | 15 | 20 | 30 |
| 0.0 | 510 * | - | - | - | - |
| 0.75 | - | 500 | 470 | 430 | 400 |
| 1.00 | - | 480 | 454 | 410 | 390 |
| 1.50 | - | 430 | 420 | 380 | 365 |
| 2.00 | - | 400 | 390 | 360 | 352 |

\* All values are given in kPa.

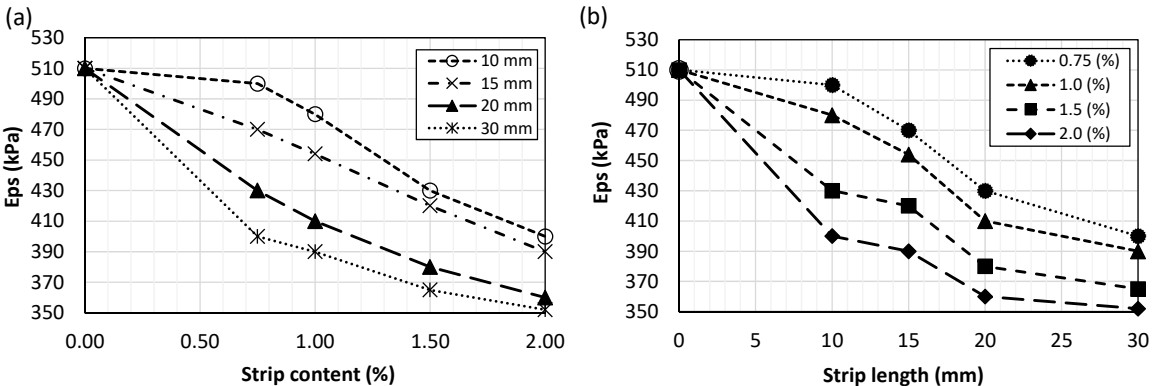

**Figure 12.** Variation of secant elastic modulus (Eps) of the strip-reinforced cemented specimen (2% cement and different strip contents). (**a**) $E_{ps}$—strip content (%); (**b**) $E_{ps}$—strip length (mm).

Another factor that can be used to check the effect of strips on the composite is the deformation strain index (DSI) [43]:

$$DSI = \frac{\varepsilon_f - \varepsilon_p}{\varepsilon_p} \quad (1)$$

where $\varepsilon_f$ is the strain at the destruction stage (final test stage), $\varepsilon_p$ is the strain at peak strength and DSI is the destruction strain index.

Table 8 presents the values of DSI and Figure 13 shows the DSI variation in terms of strip content and strip length.

**Table 8.** The DSI values for soil-cement-strip with 2% cement content.

| Content (%) | L (mm) | | | | |
|---|---|---|---|---|---|
| | 0 | 10 | 15 | 20 | 30 |
| 0.0 | 0.35 * | - | - | - | - |
| 0.75 | - | 0.27 | 0.33 | 0.63 | 0.75 |
| 1.00 | - | 0.39 | 0.41 | 0.65 | 0.77 |
| 1.50 | - | 0.42 | 0.45 | 0.70 | 0.80 |
| 2.00 | - | 0.45 | 0.6 | 0.73 | 1.0 |

\* Value of the DSI for soil without fiber with 2% cement content.



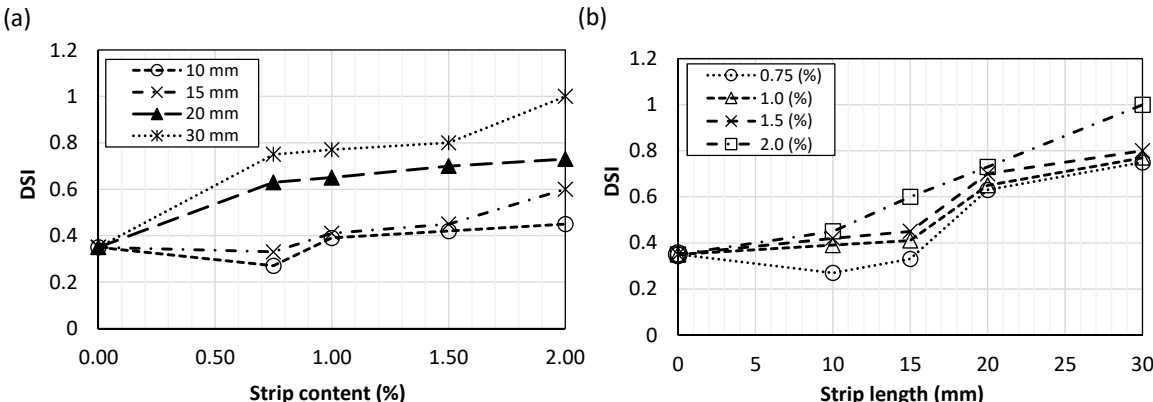

**Figure 13.** Variation of DSI (2% cement and different strip contents). (**a**) DSI-strip content (%); (**b**) DSI-strip length (mm).

DSI values showed an increase with the strip content as well as with the length when compared to the value of the soil with 2.0% cement without strips. A small initial decrease considering the strip content of 0.75% for the lengths of 10 and 15 mm was evidenced. However, in general, the value of the DSI increased for the different contents and lengths. For L = 20 mm (1.5%; 2.0%), the DSI value increased about two-fold. For L = 30 mm (2.0%), the DSI values increased about three-fold. The DSI shows the ductile response of the strips reinforced-cemented soil in comparison with unreinforced cemented soil mixture [17,43].

Tables 9 and 10 present the shear stresses at the peak obtained for each specimen and the shear strength parameters found for each soil-cement-strip envelope, respectively. Comparing the values obtained in Table 10 with the shear strength parameters obtained for the lateritic sandy soil with a 2% cement addition (23.7 kPa cohesion and 46.5° friction angle), it is possible to notice that all the analyzed inclusions showed improvement in at least one of the soil shear strength parameters, either in cohesion or in the friction angle. Only three of the analyzed inclusions presented a reduction in the cohesion parameters, whereas in relation to the friction angles obtained, eight of the sixteen analyzed inclusions presented decreases in this parameter.

**Table 9.** Normal and peak shear stresses: the soil-cement-strip (2% cement).

| %/L (mm) | 10 | | 15 | | 20 | | 30 | |
|---|---|---|---|---|---|---|---|---|
| | $\tau$ (kPa) | $\sigma$ (kPa) | $\tau$ (kPa) | $\sigma$ (kPa) | $\tau$ (kPa) | $\sigma$ (kPa) | $\tau$ (kPa) | $\sigma$ (kPa) |
| 0.75 | 60.0 | 30.6 | 98.9 | 30.6 | 93.8 | 30.6 | 44.2 | 30.6 |
| | 69.7 | 61.1 | 123.1 | 61.1 | 103.5 | 61.1 | 80.9 | 61.1 |
| | 158.3 | 122.2 | 173.1 | 122.2 | 143.5 | 122.2 | 157.1 | 122.2 |
| 1.00 | 65.2 | 30.6 | 79.6 | 30.6 | 44.6 | 30.6 | 60.0 | 30.6 |
| | 109.2 | 61.1 | 110.2 | 61.1 | 84.6 | 61.1 | 92.3 | 61.1 |
| | 180.7 | 122.2 | 193.0 | 122.2 | 143.5 | 122.2 | 153.6 | 122.2 |
| 1.50 | 79.6 | 30.6 | 67.0 | 30.6 | 71.0 | 30.6 | 64.0 | 30.6 |
| | 78.0 | 61.1 | 110.2 | 61.1 | 78.6 | 61.1 | 90.0 | 61.1 |
| | 126.6 | 122.2 | 175.5 | 122.2 | 163.4 | 122.2 | 149.6 | 122.2 |
| 2.00 | 81.0 | 30.6 | 65.2 | 30.6 | 84.1 | 30.6 | 79.2 | 30.6 |
| | 107.9 | 61.1 | 109.2 | 61.1 | 104.0 | 61.1 | 95.7 | 61.1 |
| | 193.0 | 122.2 | 163.4 | 122.2 | 114.6 | 122.2 | 110.5 | 122.2 |

$\tau$ = shear stress; $\sigma$ = normal stress; L = strip length.

**Table 10.** Shear strength parameters: the soil-cement-strip (2% cement).

| Strip Length (mm) | Inclusion (%) | c (kPa) | $\phi$ (°) |
|:---:|:---:|:---:|:---:|
| | 0.75 | 15.7 | 48.4 |
| 10 | 1.00 | 29.5 | 51.3 |
| | 1.50 | 55.3 | 28.9 |
| | 2.00 | 38.4 | 51.3 |
| | 0.75 | 73.9 | 39.0 |
| 15 | 1.00 | 38.2 | 51.4 |
| | 1.50 | 34.4 | 49.4 |
| | 2.00 | 38.1 | 46.2 |
| | 0.75 | 73.8 | 29.2 |
| 20 | 1.00 | 15.2 | 46.7 |
| | 1.50 | 28.6 | 46.7 |
| | 2.00 | 78.8 | 17.2 |
| | 0.75 | 6.1 | 51.0 |
| 30 | 1.00 | 29.4 | 45.5 |
| | 1.50 | 34.2 | 43.2 |
| | 2.00 | 71.8 | 38.6 |

c = cohesion; $\phi$ = effective friction angle.

In general, the inclusion of 2.0% of 10 mm long strips and 1.0% of 15 mm long strips proved to be more advantageous since it showed significant increases in both cohesion and friction angle. The cohesions values obtained were 38.4 kPa in the first case and 38.2 kPa in the second case, presenting increases of 62% and 61.2% in relation to the soil-cement composite and friction angles of 51.3° and 51.4° resulting in increases of 10.3% and 10.5%, respectively.

Results from Table 9 show an important characteristic of the soil-cement composite. This material may become heterogeneous. In some cases, adding strips to a cemented matrix made the behavior of the material unpredictable. The failure envelopes of some of the specimens molded with the lateritic sandy soil, added with cement and strips, were not consistent with the expected increase in shear strength. This may be related to a certain increase in the confining tension, as can be seen in the envelopes of the soil composites-strip-cement for inclusions of 10 mm strips in amounts of 0.75% and 1.5% and for strips of 20 mm in length in amounts of 1.5% and 2.0%.

Additionally, the material presented an altered rupture plain (Figure 14a). At the failure, the cemented matrix aggregates the strips in order to create a heterogeneous material, for which tensile strength and the rupture plane will depend on the distribution of the strips, especially on the number of strips that will be requested during the rupture and the way in which the efforts will be distributed. The analysis of the samples after the rupture indicated that only a small percentage of the included strips deformed definitively (showing folds), which occurred due to the efforts applied during loading or during compaction of the specimens (Figure 14b).

(a)     (b)

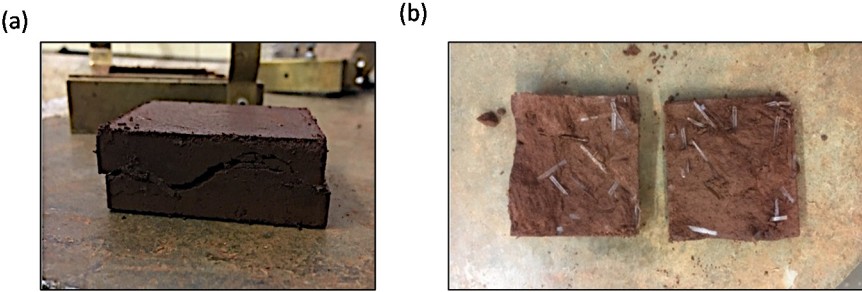

**Figure 14.** Direct shear tests. (**a**) The rupture plane; (**b**) random strip positions inside the specimen after failure.

In a more detailed analysis of the shear strength parameters, it can be seen that the friction angles and cohesions are not consistent with the expected values of a soil, so that the cement added to the analyzed composites leads to characteristics of cemented matrices, which is why the parameters obtained are so high. These factors are manifested in a more evident way for greater additions of cement or in cases in which the addition of strips occurred. Few studies have evaluated the effects of the composite formed by soil, cement, and strips. However, the results obtained are in accordance with the results of Tang et al. [17] and Olgun et al. [18].

## 4. Conclusions

This study evaluated the effect of PET strips on the mechanical properties of a cement-treated lateritic sandy soil. From the results obtained in this study, the main conclusions can be drawn:

- Regarding the uniaxial strength, all analyzed cases showed an increase in the soil strength due to the inclusion of fibers, regardless of the length and percentages in which they were included;
- Results from direct shear tests in soil-cement composites showed that in all analyzed cases the addition of cement was effective in increasing the shear strength of the soil, increasing both the cohesion parameter and the friction angle in large proportions, even when small amounts of cement have been added;
- As for the addition of fibers to the soil-cement composites, the pronounced effect occurred in increasing soil cohesion, often presenting a decrease in the friction angle. The inclusion of strips also provided a more ductile behavior to the material, presenting greater deformations with lower stress peaks;
- Results showed that the inclusion of recycled strips in soil-cement can provide a material with high strength, ductility, and a highly sustainable alternative and;
- In general, the fibers and cement addition the lateritic sandy soil mixture proved to be an excellent option for increasing the strength and deformability of the analyzed natural soil, showing high potential for applications of these materials in the civil construction industry.

**Author Contributions:** Conceptualization, M.R.S., P.C.L., and N.d.S.C.; methodology, M.R.S., P.C.L., N.d.S.C., R.A.R., and H.L.G.; formal analysis, M.R.S., P.C.L., N.d.S.C., R.A.R., and H.L.G.; investigation, M.R.S., P.C.L., H.L.G., R.A.R.; resources, M.R.S., P.C.L., N.d.S.C., R.A.R., and H.L.G; writing—original draft preparation, M.R.S., P.C.L., N.d.S.C., R.A.R., and H.L.G.; writing—review and editing, M.R.S., P.C.L., and N.d.S.C.; visualization, M.R.S., P.C.L., N.d.S.C., R.A.R., and H.L.G.; supervision, P.C.L. and N.d.S.C.; project administration, M.R.S., P.C.L., N.d.S.C., R.A.R., and H.L.G. All authors have read and agreed to the published version of the manuscript.

**Funding:** This research received no external funding.

**Acknowledgments:** The author are very thankful to Capes/Print program and PROPG/UNESP.

**Conflicts of Interest:** The authors declare no conflict of interest.

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
