# Peer review of "Effect of Recycled Polyethylene Terephthalate Strips on the Mechanical Properties of Cement-Treated Lateritic Sandy Soil"

_sustainability, doi:10.3390/su12239801_

Round 1
Reviewer 1 Report
The authors have studied the impact of the inclusion of polymeric strips of polyethylene terephthalate and cement on the mechanical strength of a sandy soil by conducting unconfined compression and direct shear tests. The subject is relevant to the scope of the journal and the paper is well organized. It is recommended for publication after addressing the following comments.
- The literature review can be improved by presenting all similar studies and comparing them with the work done in the manuscript.
- Please calculate and present percentage difference of mechanical strength parameters before and after the inclusion of polyethylene terephthalate strips.
- It is recommended that the authors present soil Poisson's ratios before and after the inclusion of polyethylene terephthalate strips.
- The reviewer recommends that the authors quantify the cost reduction for “low-cost” mentioned in Conclusions: “Results showed that the inclusion of recycled strips in soil-cement can provide a low-cost material“.
Reviewer 2 Report
General comments:
- Use „ lateritic sandy soil” throughout the article.
- Use (-) instead (x) throughout the article.
- Make a decision: peak, destruction, brake or rupture stages and apply consistently throughout the article.
- The methodology of samples preparation for direct shear testing should be described in detail. Generally, we have a problem with reaching 100% compaction. For sand-cement composites, the time from sample preparation to testing is very important. The use of maximum normal stress 120 kPa should be commented.
- Use “maximum dry density” instead of “specific maximum density”.
Detailed comments:
Line 4. Better “…Lateritic Sandy Soil.”
Line 43 and 44. ( written by Laura Parker in 2018; does not appear in References).
Line 133. … in Figure 1.
Line 237. Must be corrected.
Line 259. Better “cement content” than “cement inclusion”.
Line 346. Should be corrected. All values are given in kPa.
Line429. The table shows not only shear stresses but also normal stresses.
Comments on figures:
Fig. 2. The figure shows four curves but only two are described. The vertical axis should be named “Unconfined compression stress”.
Fig. 3. The two curves should be shown in one figure, not two figures. The strains for the maximum stresses should be marked and difference should be 1.5%.
Fig. 4. The figure is illegible and should be separated into two. One for dilative and others for contractive behaviour.
Round 2
Reviewer 1 Report
The authurs have addressed my comments. The paper is recommneded for publication.
Author Response
Thank you for your comments. We performed a spell check in order to minimize spelling errors.
(The corrections are highlighted in blue color in the text).